# ANYVIEW: FEW SHOT PERSONALIZED VIEW TRANSFER

## ABSTRACT

Fine-tuning generative models for concept driven personalization have witnessed tremendous growth ever since the arrival of methods like DreamBooth, Textual Inversion etc. Particularly, such techniques have been thoroughly explored for style-driven generation. Recently, diffusion models have also demonstrated impressive capabilities in view synthesis tasks, setting the foundation for exploring view-driven generation approaches. Motivated by these advancements, we investigate the capacity of a pretrained stable diffusion model to grasp "what constitutes a view" without relying on explicit 3D priors. Specifically, we base our method on a personalized text to image model, Dreambooth, given its strong ability to adapt to specific novel objects with a few shots. Our research reveals two interesting findings. First, we observe that Dreambooth can learn the high level concept of a view, compared to arguably more complex strategies which involve fine-tuning diffusions on large amounts of multi-view data. Second, we establish that the concept of a view can be disentangled and transferred to a novel object irrespective of the original object's identity from which the views are learnt. Motivated by this, we introduce a learning strategy, AnyView, which inherits a specific view through only one image sample of a single scene, and transfers the knowledge to a novel object, learnt from a few shots, using low rank adapters. Through extensive experiments we demonstrate that our method, albeit simple, is efficient in generating reliable view samples for in the wild images. Code and models will be released.

## 1 INTRODUCTION

In the recent times, diffusion models Ho et al. (2020); Song et al. (2020); Rombach et al. (2022) have shown excellent results for high quality image generation. They have been shown to have impressive understanding of high level concepts of art stylesShah et al. (2023) and object level details. Additionally, these models offer controllability in the form of conditioning, with text being the most common form of conditioning. Several text controlled approaches like DreamBoothRuiz et al. (2023), textual inversionGal et al. (2022) have allowed personalizing diffusion models on an object level. These methods have further progressed to learn abstract concepts such as artistic style Wang et al. (2023a); Shah et al. (2023), shedding light on the fundamental problem of whether diffusion models is capable of learning other abstract concepts as well. In this work, we attempt to develop a finer understanding on this problem by trying to learn the concept of a visual view. While both NeRF based methods Mildenhall et al. (2021); Barron et al. (2021); Yu et al. (2021); Deng et al. (2022); Niemeyer et al. (2022) and diffusion based approaches Gu et al. (2023); Tseng et al. (2023); Ye et al. (2023) typically rely on three-dimensional priors requiring extrinsic and intrinsic camera poses for reliable view generations, our goal is to do so without.

Interestingly, the human brain does not require any camera poses to perceive the viewpoint of an object in a photo. It can be analogous to a model which has been trained on a vast number of different instances of data which allows it to provide an estimate of the view. We then question whether pretrained diffusion models trained on a virtually exhaustive amount of data is already capable of understanding the viewpoint of any object through visual cues alone such as its spatial relations to other objects in background? In order to answer this, we conduct a simple experiment with text to image personalised models (in our case DreamBooth Ruiz et al. (2023)). Utilizing DreamBooth's setup is appealing for two reasons. First, DreamBooth achieves high fidelity to the given subject context with as few as 3-4 samples. Second, it leverages the extensive pretraining of

the underlying diffusion model to generate new concepts. We synthetically generate several samples of chair views with different backgrounds through in-painting Yu et al. (2023) and assign a unique ID to each view. We observe that the unique ID not only faithfully binds to the context of the view but also reliably disentangles the context from a random object distribution in the diffusion model's knowledge space, as shown in Figure 1. Moreover, as shown in Figure 1 as well, just by prompting a stable diffusion model with "a top of dog ..." versus "a top view of dog ..." yields completely different results in terms of viewpoint. The latter indeed generated a top view, hinting that a diffusion model greatly understands the concept of the keyword "view".

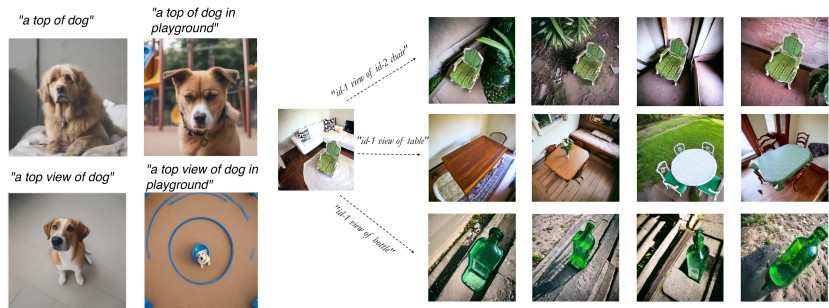

Figure 1: **Impact of transferring a view concept to different object distributions in diffusion's pretrained space.** On the left; we show that a stable diffusion model itself already recognizes the meaning of the word "view". On the right; we use Dreambooth with view id instead of object id. Row 1 shows that a particular view of a chair can be retrieved with an unique id assigned for the concept. Row 2 and 3 show that the abstract concept of this view is learnt in disentangled manner which allows reliable reconstruction of other object in the same view point.

Motivated by these findings, we hypothesize that diffusion models can learn high-level concepts such as view, analogous to style Wang et al. (2023b); Sohn et al. (2024); Shah et al. (2023) and gender Wang et al. (2024), in a disentangled manner from object identity. We then formulate the task of *personalized view transfer* in the following manner: given 2D image(s) of an object's view, a pre-trained diffusion model learns the specific view as a high-level concept and and can transfer the knowledge of the view by generating a novel object (both specific or not) in the same view.

Leveraging DreamBooth's findings, it is possible to learn the view/scene concept and novel object concept with minimal 2D data and no pose metadata. By combining these models, the view and scene can be transferred to the novel object. However, in relation to our work, we have observed that Dreambooth is not capable of learning more than one id (whether view or object). Specifically, we find that Dreambooth will forget a previous id that it has learnt if we are to train it with another id subsequently. We refer to this as the "forgetting" problem, and outline how we resolve it in the list of contributions below:

1. We provide empirical evidence that a view, which describes the spatial relationship of an object with its surrounding in a 3D space, can be treated as a concept to train contextually personalized diffusion models analogous to the concept of style.

2. We establish that the learned view concept is identity-independent and transferable to novel objects with different geometries. We derive a learning strategy as follows: first, learn the view concept; second, learn the user-specified object; and finally, merge the two to generate novel views of out-of-distribution images. To avoid the forgetting problem, we use LoRA Hu et al. (2021) to learn the view, object, and merged concepts *separately*.

3. Tapping into the few-shots nature of Dreambooth, our method operates under a few-shots constraint, using as few as a single sample of an object to learn a view LoRA and 3-4 samples to learn a novel object LoRA, thereby avoiding extensive pretraining on multi-view data. We call our method AnyView (transferring virtually any possible view). To avoid confusion, we note that the original Dreambooth does not utilize LoRA. AnyView does so in order to create separate view and object entities that can be further merged (Figure 2).

4. We provide extensive ablations of AnyView for several uses cases using in the wild images and benchmark our method on widely used DTU dataset for the task of novel view synthesis to show the efficacy of the view transfer.

## 2 RELATED WORKS

### 2.1 NOVEL VIEW SYNTHESIS USING IMPLICIT NEURAL REPRESENTATIONS

The domain of novel view synthesis has recently centered around implicit neural radiance fields (NeRFs) Mildenhall et al. (2021). Contemporary methods often use NeRF as a backbone, typically requiring several images of a scene to generate multiple views. Recent efforts aim to achieve "NeRF-like" reconstruction with fewer images Yu et al. (2021); Roessle et al. (2022); Zhang et al. (2021); Niemeyer et al. (2022), relying on image-based feature extraction followed by end-to-end training with some 3D supervision. For example, PixelNeRF Yu et al. (2021) uses a CNN-based feature extractor with differentiable un-projection of a feature frustum from input images. Generalizable NeRF Transformers Wang et al. (2022) replace ray tracing with a transformer block that aggregates multi-view image features. Improvements like Mildenhall et al. (2022) enhance robustness to noise and quantization errors. However, these methods often require per-scene training or extensive multi-view data for generalization.

### 2.2 NOVEL VIEW SYNTHESIS USING DIFFUSION MODELS

Text-to-image models like DALLE Ramesh et al. (2022), Latent Diffusion Rombach et al. (2022), and Imagen Saharia et al. (2022) excel at generating high-resolution, realistic images in a zero-shot manner. Unlike text-to-3D models, view synthesis from a few images must preserve the visual features of the input. Diffusion based models have been proposed for object centric view synthesis; a task that is usually divided in two primary stages that includes training a 3D aware diffusion model followed by transferring the 3D consistent information to the input scene given. The score distillation sampling in Poole et al. (2022) uses a 2D diffusion model as a prior for a parametric image generator which then is used to optimise a NeRF model for a text to 3D task. Methods like DiffRF Müller et al. (2023) noise and de-noise a radiance field followed by volume rendering for realistic object-centric views but require ground truth radiance fields and are computationally expensive, limiting resolution. Other works such as Chan et al. (2023), Xiang et al. (2023), Watson et al. (2022) involve explicitly incorporating 3D geometric priors into diffusion models to generate 3D synthesis. While GenVS depends on evaluating models on one scene category like table or fire hydrants on a single run, zeroNVS Sargent et al. (2023) can process multiple categories for evaluation in a single model. However the visual results often are far more compromised and look blurred with zeroNVS.

Considering that diffusion models are trained on vast amounts of web data Rombach et al. (2022), many existing works employ finetuning strategies to harness the trained latent space. Dreambooth Ruiz et al. (2023), for example, generates high-fidelity images of a specific object with only a few sample images. This has led to focused research on diffusion models for novel view synthesis. Methods like ViewNeTI Burgess et al. (2023) use camera viewpoint parameters, Ri and a scene-specific token, $S_o$ to predict latents in the CLIP text space, employing neural mappers to produce word embeddings. Nerdi Deng et al. (2023) uses image captions and word embeddings extracted via textual inversion for the diffusion network. However, most methods rely on 3D priors, such as camera extrinsics and intrinsics, leaving room to explore whether diffusion models can understand 3D scenes without explicit 3D priors.

### 2.3 UNSEEN DOMAIN GENERALIZATION USING LOW-RANK ADAPTATION (LORA)

Low Rank Adaptation Hu et al. (2021) is an uniquely intuitive way of reducing the effective number of trainable parameters, when finetuning large scale models, such as Stable Diffusion Rombach et al. (2022). LoRA based finetuning methods only learn a trainable matrix of a very low intrinsic rank, which makes the training and storage of these weights efficient. LoRA based finetuning poses other advantages: pre-trained models can be equipped with any set of LoRA adapted weights for different domain adaptation tasks. Methods such as Shah et al. (2023) Huang et al. (2023) Xia et al.

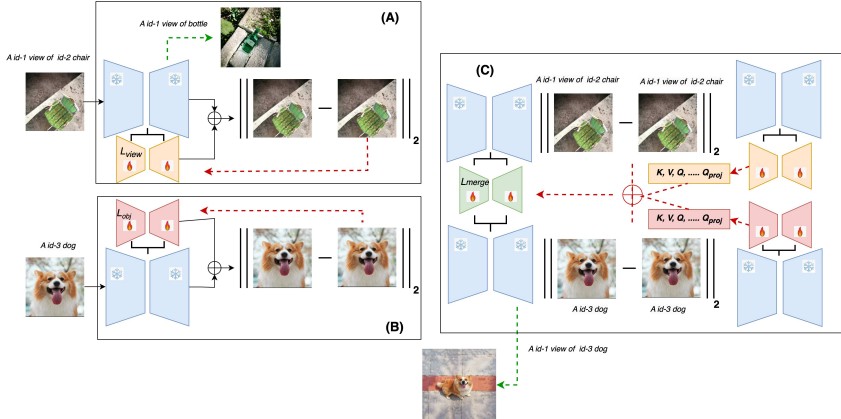

Figure 2: **Proposed approach for view transfer to unseen objects.** The blue frozen model is a SDXL Podell et al. (2023) pretrained model. **(A)** We train the LoRA adaptor of the diffusion model to learn a concept of view from an image using a text prompt with two unique identifying tokens, one for the view and the other for preserving the visual concept of the original object. The green dashed line shows an inference where we simply apply the unique token of view to the specific view of a bottle. **(B)** The LoRA adaptor of the diffusion model here learns the visual content of the new images of a novel object (dog), using an unique token for it. **(C)** In this stage, we merge the object and view LoRA adaptors with ZipLoRA. The red dashed line show the flow of gradient during back propagation.

(2024)Renduchintala et al. (2023) explore the possibility of using multiple sets of LoRA weights and efficient ways of combining these weights for the models to use. With this development as prior to our research, we are posed with a possibility that such methods, which learn finetuned concepts in a manner disjoint from the original pre-training, may also be able to learn view and object concepts. We make reasonable efforts to explore these research gaps.

## 3 METHODOLOGY

We dedicate this section to a discussion of the methodologies we adopt in our work. Our method involves low rank adapted finetuning Hu et al. (2021) on a stable diffusion XL model Podell et al. (2023) to learn object and view concepts and then eventually combine these concepts together to transfer the learnt view to the user specified object. The training strategy of AnyView is divided into three stages as shown in Figure 2. In stage-A, we finetune the base stable diffusion weights with view specific LoRA adaptors to learn the concept of view with a single image sample for the corresponding view. In the second stage-B, we train object specific LoRA adaptors, which can learn the visual attributes of the novel object from 3-4 samples of the object. Both stage-A and B LoRAs are finetuned following the Dreambooth method. Finally, in stage-C, we merge the two concepts adaptively with guidance from the previously trained LoRA adaptors. In all the three stages we keep the base stable diffusion model weights frozen updating only the key, query, value and their projections in attention modules following the LoRA Hu et al. (2021) based finetuning literature.

### 3.1 PROBLEM STATEMENT

Given a set of images $\Phi_v = \{x_i\}$ where $i \in [1, N]$, $N$ being the number of samples each representing the same specific view, $v$, and a pre-trained generative model, $D_\theta$, $\theta$ being the base model parameters, we finetune $D_\theta$ on $\Phi_v$ by adapting to a set of weights with a lower intrinsic rank, to that of the original weights $\theta$ Hu et al. (2021). Our assumptions are as follows: views exists in the latent space of generative models as high level concepts, and these can be learnt in a manner, disentangled from other concepts. The objective is to train $D_\theta$ on the high-level concept of camera view, $v$, so that the updated model low rank weights, $L_v\{D_\theta\}$, can be transferred to the unknown images, $\Phi_o$, of a novel object, $o$, generating images, $\hat{\Phi}_o$, that share close fidelity to $\Phi_o$ while preserving the concept

$v$. Our hypothesis is that $v$ is learnt in a disentangled fashion from the object's identity. Therefore, $v$ and $o$ can eventually be merged together to generate sample image adhering to both concepts.

## 3.2 FINETUNING THE STABLE DIFFUSION WITH A COMBINATION OF DREAMBOOTH AND LOW RANK ADAPTER

We follow the DreamBooth finetuning protocol to personalize the SDXL Podell et al. (2023) base model to a specific view or an object. In other words, we simply finetune a diffusion model with text prompts like "A [object id] dog", except that while learning the view, we assign an additional specific [view id] for the reference view object in the form of "A [view id] view of [object id] object-name". Further, in our experience with keeping all the layers of the diffusion network unfrozen for training the concept of view and object sequentially, we experience the forgetting problem for the concept which was learnt first Smith et al. (2023). Therefore, we utilize LoRA Hu et al. (2021) and update only specific LoRA layers instead of updating the entire SDXL model as shown in Figure 2, training LoRAs for the view and object concept separately. LoRAs hence act as expert models that can be merged with an added benefit of computational efficiency. The only crucial difference in the two trainings is that the object training involves a few (3-5) training images, while the view concept training is done on only one view image. Additional implementation details can be found in Hugging Face's diffusers. Subsequently, upon training specific LoRA weights, $L_v$ and $L_o$, for the view and object concept respectively, we find that just simply performing a linear combination of $L_v$ and $L_o$ results in ambiguous artifacts in the generated image. We hence merge them using the popular ZipLoRA Shah et al. (2023) to transfer the concept of the view to the object, which works reasonably well in our experiments. For additional details on LoRAs, merging and loss functions used can be found in Section A.1, A.2 and A.3 respectively in the appendix.

## 4 EXPERIMENTS

### 4.1 COMPARISON ON DATASETS

**Dataset and evaluation protocol**: Given the unique nature of the personalized view transfer task, we had to come up with benchmarking strategies that can be useful for evaluating our method. For this purpose, we apply AnyView to the novel view synthesis (NVS) task on the DTU MTS Aanæs et al. (2016) dataset, even though AnyView is meant for *view transfer and not view synthesis*. In order to get the novel views on the test set, we first train the reference view LoRAs with a scene from the training set, selected randomly for fairness, which contains the views that needs to be transferred. Then, we train the object LoRAs for the test set object class, selecting 4 images in random. Finally, we merge the respective view LoRAs with object LoRAs to generate the final image. We compare the generated image against the ground truth images corresponding to various camera views of different objects provided in DTU. We use the 15 test scenes and 6 evaluation scenes from the literature Burgess et al. (2023); Deng et al. (2023); Yu et al. (2021). To calculate unbiased SSIM scores Wang et al. (2004), we generate segmentation masks for both original and generated scenes using SAM Kirillov et al. (2023), with the caption input to SAM being the class name of the object.

**Experimental Setup**: All experiments are performed on the SDXL v1.0 base model Podell et al. (2023), using the default settings of ZipLoRA Shah et al. (2023) for finetuning. Input images are resized to $1024 \times 1024$ and the batch size is set to 1 for all stages. We finetune AnyView for 1000 iterations in the view and object training stages, and for 100 iterations in the final merging stage. The default SGD optimizer with a constant learning rate of $5e - 5$ is used. The base model weights and text encoders remain frozen, updating only the LoRA layers (query, key, value, and their projections in self and cross attention modules). The LoRA rank is set to 64, and the cosine multiplier $\lambda$ is 0.01. Unique identifiers follow the DreamBooth protocol. However, we do not use any geometric augmentations like random flipping or cropping as it changes the definition of a view. All experiments are done on a single H100 gpu with 80 gigs of memory. Training the view and object LoRA takes the same time as would the normal finetuning of DreamBooth, roughly 30 mins on our hardware. Both the LoRAs can be finetuned in parallel if a second gpu is available. The merging of the two LoRAs require 15 mins for a single view transfer.

**Baselines**: We compare AnyView with two few shot state-of-the-art benchmarks PixelNerf Yu et al. (2021) and ViewNeti Burgess et al. (2023) (both of which also perform in a single image setting). Both of these methods have competitive advantage over AnyView as they underwent prior training on the *full* training set of DTU MVS dataset, unlike our few shot approach. We also compare our method to NeRDi Deng et al. (2023), which uses depth maps to regularize the 3D geometry, which were not available to AnyView.

**Results**: The quantitative benchmarks are based on three widely used metrics, SSIM Wang et al. (2004), PSNR and LPIPS Zhang et al. (2018). As seen in row-6 in Table 1, AnyView achieves the best performance on LPIPS and PSNR, and the second best performance on SSIM, losing to Pixel-NeRF Yu et al. (2021) slightly in spite of the competitive advantage PixelNeRF has. Additionally,

Table 1: Comparison for novel view synthesis on DTU dataset. The best scores are in bold and the second best are underlined.

| Methods | LPIPS ↓ | SSIM ↑ | PSNR ↑ |
|---|---|---|---|
| NeRF Mildenhall et al. (2021) | 0.703 | 0.286 | 8.000 |
| PixelNeRF Yu et al. (2021) | 0.515 | **0.564** | 16.048 |
| SinNeRF Xu et al. (2022) | 0.525 | 0.560 | 16.520 |
| NerDi Deng et al. (2023) | 0.421 | 0.465 | 14.472 |
| ViewNeTI Burgess et al. (2023) | 0.378 | 0.516 | 10.947 |
| AnyView (ours) | **0.375** | 0.563 | **26.587** |

AnyView outperforms SinNeRF Xu et al. (2022); even though SinNerf uses single image for novel view synthesis, it has an unfair advantage of using depth maps and geometric pseudo labels for regularization, which are not available to AnyView. Furthermore, we agree with the remarks made by Deng et al. (2023) Burgess et al. (2023) that the reconstruction based metrics are not appropriate for few shot view setting as the generative models rely on hallucinating the unseen regions of the images. These metrics tend to rely on averaging over multiple views than providing a score for visually reliable views. This is an area where NeRF based methods naturally excel at.

**Qualitative Results**: The qualitative performance of AnyView on the DTU MVS Dataset is shown in Figure 3, and results on natural images from the DreamBooth Dataset are in Figure 4. We use class names to identify objects in prompts, a *unique-id* for recognizing the view, and another *unique-id* for recognizing the object in reference view. The object LoRA is trained with 3-4 reference images of new object samples and merged with the view LoRA. As seen in Figure 4, reconstructions for natural images appear visually better. We attribute this to stable diffusion being pretrained on millions of

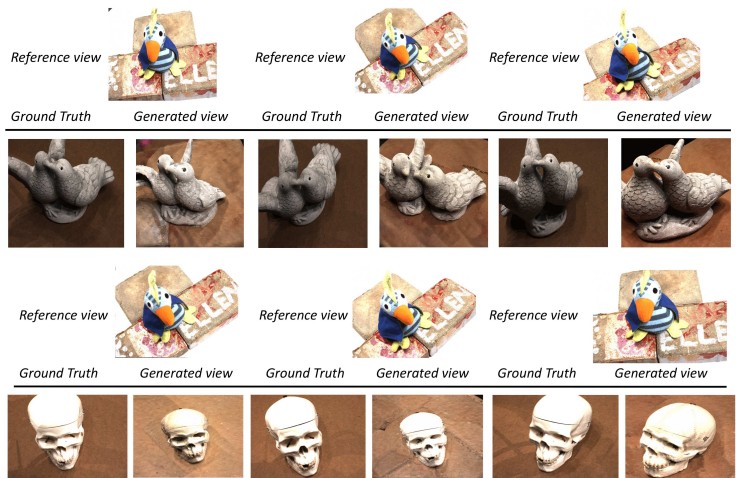

Figure 3: **View Transfer on DTU MVS Dataset.** Given the reference view to train view adaptor and image samples of the novel object (statue in row-1 and skull in row-2) we compare the synthesized views with original ground truth views available.

natural images, enhancing its hallucination ability for such images. Unlike other methods, AnyView does not learn the DTU dataset distribution for subsequent view reconstructions as its goal is to

facilitate view transfer in the few shot settings. For additional qualitative results on DTU dataset and comparisons, please refer to Figure 12 and Figure 14 in  in appendix.

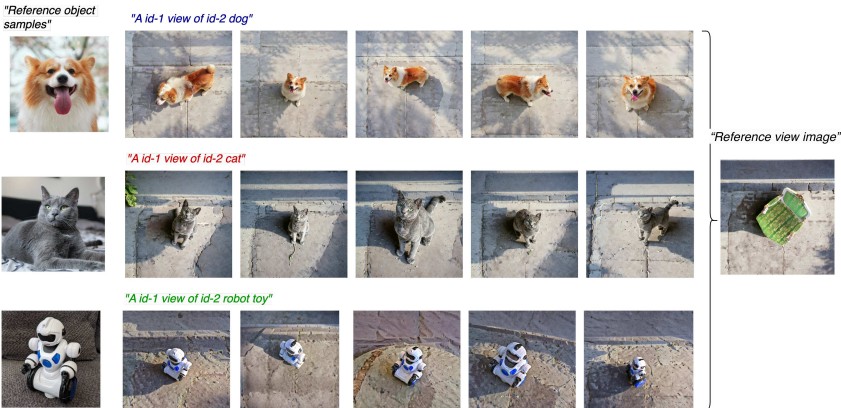

Figure 4: **View Transfer on DreamBooth Dataset.** Given reference view image of a chair from top view, we generate top views of different objects like dog, cat, robot-toy etc. AnyView reliably hallucinates the top views (row-1 and row-3) despite the difference in the structure of the reference view object and novel object.

**Comparison with Zero123**: Zero123 Liu et al. (2023) generates an image of an object when given a reference image of the same object and a desired pair of rotation and translation, $(R, T)$. Although primarily, Zero123 is performing an NVS task, it can be utilized for the view transfer task. Specifically, we manually estimate the $(R, T)$ needed to get the best view transfer possible by also trying a few transformations. Following this, we believe it is meaningful to compare AnyView to Zero123. In our experiments, we have observed that Zero123 often produces distorted objects that share low fidelity with the original object. For example, the details in the car are mangled in Figure 5, and the cars in row 3 and row 5 are barely consistent with the reference object. On the other hand, AnyView maintains high fidelity to the original object while remaining faithful to the reference view.

## 4.2 ANALYSIS

**Is the view actually learnt?** We conduct an ablation study where we train view LoRAs with different reference views to see if the concept of a view is learnt, and if it can be transferred to different objects generated randomly from diffusion space with varying degrees of complexity. The results in Figure 6 provides evidence, that referring a view with a specific ID disentangles it from the reference object's identity. Furthermore, the concept is learnt well enough to synthesize complex objects like a house or boat in the specified view.

**Why do we need one view per LoRA?** So far, all our experiments train one LoRA per view. In our experiments in Figure 7, we attempt to train a view LoRA with multiple view concepts. In this setting, an unique identification token is assigned to each view but the unique identification token with the corresponding object concept remains the same for each view. We train the network for 5, 10 and 15 views at a time. It can be seen in Figure 7 that there are some outliers in all three settings. By outliers, we do not refer to the pose of the object in the image. In row 1 of Figure 7, it can be seen clearly that in the first and last sample the camera view wrt to the slab on ground has changed. Even with a larger training set where we created multiple images for a specific view by inpainting with different backgrounds, such outliers remain. We conjecture that this may be because the number of training samples required increases greatly as the number of views per LoRA increases, which defeats our original few shot motivation. As a result, without loss of generality, we restrict to one view per LoRA.

**Does AnyView work on complex objects?** Previously for training the view LoRA adaptor, we had selected a random view from the DTU MVS dataset, *"a bird toy"* and another one from the NeRF

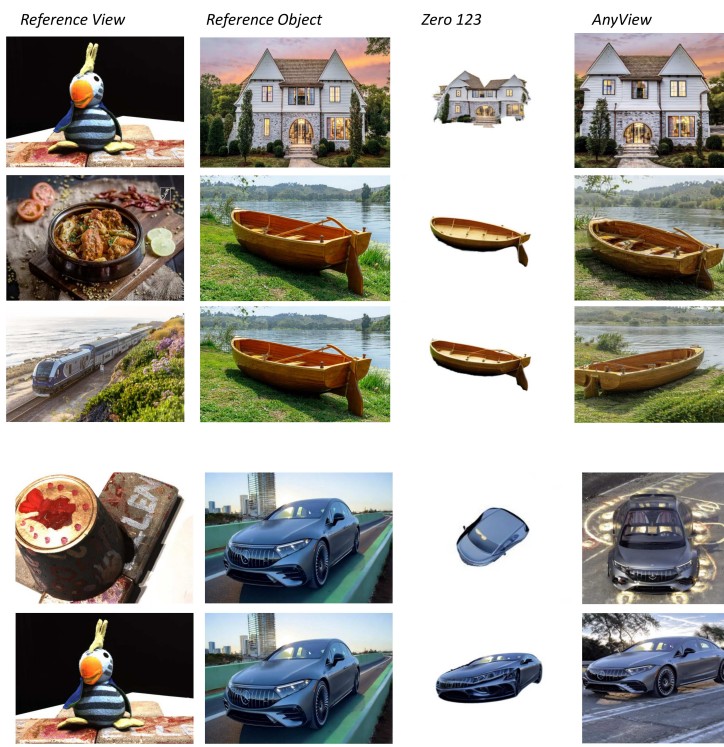

Figure 5: **Comparison of AnyView to Zero-1-to-3 Liu et al. (2023).** Our input object reference and input view reference are shown in the two leftmost columns. For Zero-1-to-3, we manually estimate the transformation in order to get the best view transfer possible by also trying a few transformations. Overall, while running Zero-1-to-3, we find that the view transfers it produces are generally of lower quality.

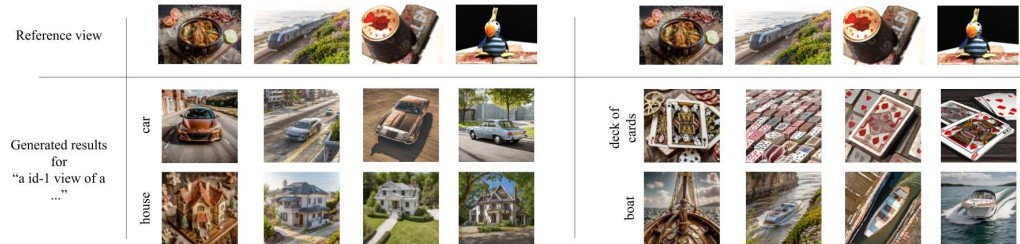

Figure 6: **Row 1** shows the reference views used while training the view LoRA. The prompt used is "a photo of *view-id* of [*reference view object name*]". The reference view is evidently accurately transferred to several different objects that are complex and geometrically very different from the reference views.

synthetic dataset, *"chair"*. Both of the objects are fairly simple to generate but the question arises whether AnyView can reconstruct different views of a complex object (like person) from inanimate simple objects. First, we evaluate if it is at all possible to generate humans through AnyView. In Figure 8, we use reference views of an athlete for view training and use the images of "Jamie Lannister", a character from the popular TV show Game of Thrones as the novel object.

It can be seen that our method can reliably generate the views of human given proper view reference images. To this end, we further evaluate our method on an extreme case of generating the top views of Jamie Lannister given a chair from the top view as the reference view and only forward facing sample of the character for object training. In Figure 9, it can be clearly seen that from a simple object like chair we can still generate views of specific characters. Although the overall

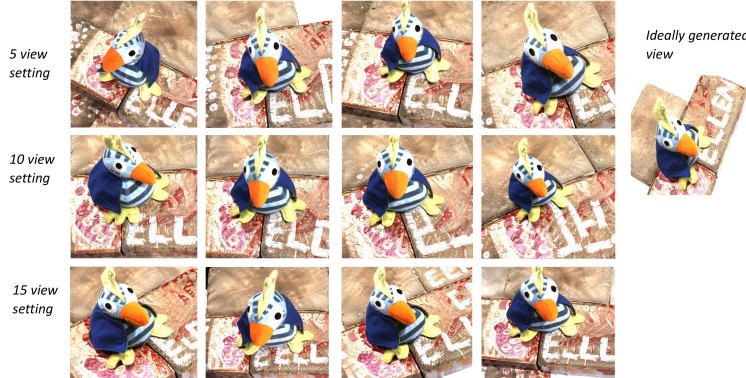

Figure 7: **Comparison of multi-view LoRAs.** We train the view LoRA adaptor with multiple views in the same model. The number of views per LoRA is varied in steps of 5,10,15. The view reconstructions in all three of the cases deviate from the reference camera view. Here, the inference was conducted on the view LoRA itself.

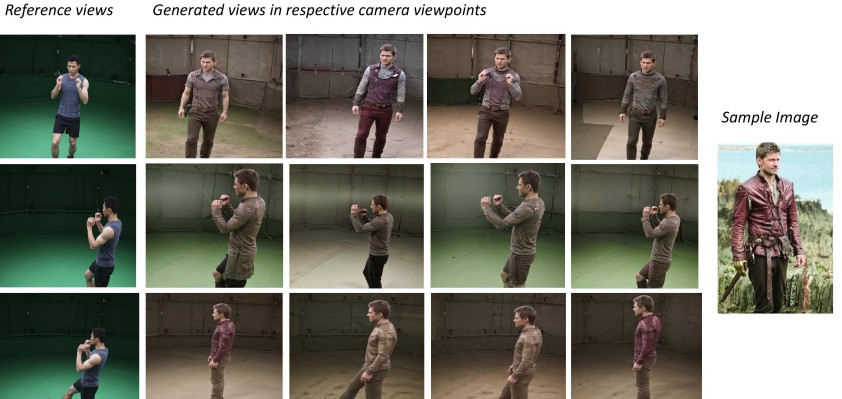

Figure 8: **View Transfer on person object.** Given the reference views of athlete on the left we generate a popular TV show character on the right. The object LoRA is trained with images of the character taken from web.

quality of generation drops a little, the results are both consistent to the view and the character itself. This demonstrates the few shot abilities of AnyView for transferring the view to complex in-the-wild images. We also explore the complexity in view generation due to multi-object images and occlusions in Figure 21 and Figure 20 in section A.7 in the supplementary section.

**Does changing the background have any effect on the view transfer?** In this section, we examine the impact of the background on the generations after merging the view with the object concept. We select four backgrounds: beach, forest, grass, and table. To add the background to the reference view object, we use stable diffusion as an inpainting model. We generate the object's mask with SAM Kirillov et al. (2023), invert it, and provide the respective background prompt (e.g., "on a table") to reconstruct the background. These images are then used for view training followed by merging with a corgi dog concept. The results are shown in Figure 10. Part **(A)** shows generations with a beach background, while parts **(B)**, **(C)**, and **(D)** show grass, forest, and table backgrounds, respectively. Complex backgrounds like a forest or table, with artifacts such as table edges or tree positions, help the diffusion model establish spatial relationships, resulting in more faithful view generations. In contrast, the grass background, lacking such artifacts, leads to variable generations. These observations support our definition of view, suggesting that camera view is estimated through other objects in the ground plane. Further details on the importance of visual cues in background generation are discussed in section A.5 of the appendix.

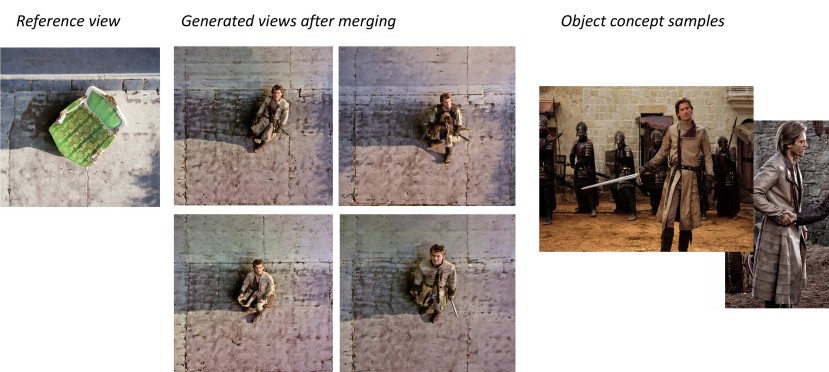

Figure 9: **Transfer of view in structurally different object.** The view LoRA trained on a chair view and object LoRA trained on a person is merged to synthesize views of the person.

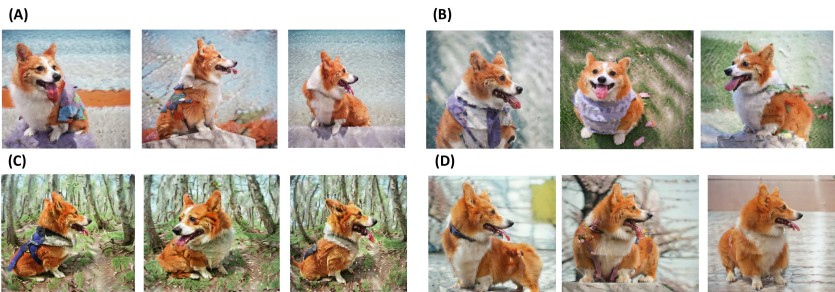

Figure 10: **Effect of backgrounds in merging the view and the object adaptors. Part(A)** uses beach in the background for view training. **Part(B)**, **Part(C)**, **Part(D)** uses grass, forest and table respectively. The generation of views are better when background has anchoring artifacts which allows the model to learn the view point of the object wrt to the objects in ground plane (e.g part (D) vs part(B).

## 5 CONCLUSION

In this paper, we report an interesting finding that it seems diffusion models are capable of capturing specific viewpoint and object concept at a high level without needing any 3D prior knowledge. Harnessing the extensive coverage of a SDXL model, we conducted LoRA style learning of view and object before merging them. It appears our proposed pipeline is quite capable of disentangling and transferring the learned view to the novel object. Ablative experiments led us to believe that the view is learned via spatial relationships to the background. AnyView requires no 3D priors or pretraining and could be a strong contribution to AI researchers seeking to transfer views with minimal overhead as each view LoRA requires only one sample per view while the object LoRA 3-4 samples.

**Potential Impacts** We acknowledge that this work has implications of generating deepfakes, however we believe that it is important to establish an understanding about the applications of generative networks for detecting fake images.

**Limitations**: Firstly, AnyView has one LoRA per view. For efficiency reasons, multiple view information present in a single LoRA can greatly reduce the amount of time taken for fine-tuning. Secondly, complex scenes involving multiple objects of interest remain a challenge for AnyView. Finally, there is a need for a more controllable generation of the background in the final image. We hope to overcome these limitations in future.

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

## A   APPENDIX

### A.1   BACKGROUND: VIEW TRAINING AND OBJECT TRAINING WITH LOW RANK ADAPTERS

In LoRA Hu et al. (2021) training, the weight updates $\Delta\theta$ to the base model weights $\theta$ where $\theta \in \mathbb{R}^{m \times n}$ can be decomposed into two intrinsic matrices which are lower in rank. Typically weights for layer $i$ is represented as $\theta^{\{i\}}$, but we drop the index notation for simplicity. If matrix $\Delta\theta$ is of size $m \times n$, it can be represented as a matrix multiplication of two matrices $A$ and $B$ of size $m \times r$ and $r \times n$ respectively, r being the intrinsic rank of $\Delta\theta$. Therefore, $\Delta\theta = A \cdot B$ where $A$ and $B$ are trainable. For inference, the weight matrix $\theta'$ can be obtained as $\theta + \Delta\theta = \theta + A \cdot B$. Let the pretrained stable-diffusion model $D$ be initialized with weights $\theta_0$. We finetune the model on distribution of $\phi_v$ with a unique view token for the view as well as a unique object token for the object. Thereby, the text condition to the model becomes "A *unique-id-1* view of *unique-id-2* [*class object*]". Given this, the view specialised weight updates $\Delta\theta_v$ can be decomposed as shown in Equation 1. Finally, the weight updates could be added to the base model weights.

$$\Delta\theta_v = A_v \cdot B_v \tag{1}$$

The weight matrix for inference can be obtained as $\theta_0 + \Delta\theta_v$. We train the object LoRA in similar manner as the view LoRA using only one object specific *unique-id* token in the prompt. Subsequently, upon training specific LoRA weights $L_v$ and $L_o$ for each concept, we merge them as laid down in Equation 2. The only crucial difference in the two training is that the object training involves a few (3-5) training images, while the view concept training is done on only one view image.

### A.2   MERGING OF OBJECT AND VIEW LoRAS

The two LoRA weight update matrices, $\Delta\theta_v$, $\Delta\theta_o$, can be merged as a linear combination of the individual weight updates. This means that the merged LoRA weights $L_{vo}$ are given as:

$$\Delta\theta_{vo} = w_v \cdot \Delta\theta_v + w_o \cdot \Delta\theta_o = w_{vo} \cdot \Delta\theta_v + (1 - w_{vo}) \cdot \Delta\theta_o \tag{2}$$

where $w_v$, $w_o$, $w_{vo}$ are scalar weights and $\Delta\theta_{vo}$ is the weight update matrix of merged LoRA. These weights can be tuned in order to gain control over the influence of concept learning $v$ and $o$. However, we observe in linear merging of the two LoRAs, the identity of the object from which the view is learnt and the unseen object's identity would either superimpose with each other resulting in concept leaks or would result in broken reconstructions as shown in Figure 11.

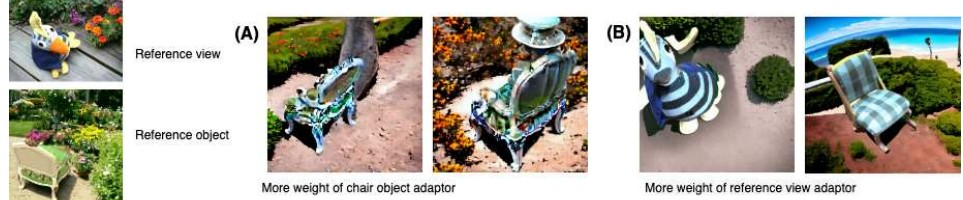

Figure 11: **The problem with linearly weighted merging.** In **part (A)** when the weight of the object adaptor is kept high we see broken reconstruction of chair. In **part (B)**, the weight of view adaptor is high resulting in concept leaks.

To mitigate this issue, we adopt the style transfer merging in ZipLoRA Shah et al. (2023) to transfer the concept of the view to the object. As an alternative to Equation 2 in appendix, the scalar constant, $w_{vo}$, can be replaced with a coefficient vector for better merging. To this end, the merging process becomes:

$$\Delta\theta_{vo} = m_o \otimes \Delta\theta_o + m_v \otimes \Delta\theta_v \tag{3}$$

where $m_o$ and $m_v$ represent coefficient vectors having the same dimensions as the corresponding $\Delta\theta$ and $\otimes$ represent an element-wise multiplication.

Each weight matrix is a linear transformation being defined by its columns. Hence, the merged LoRA would retain the available information in these columns only if the columns are being added

orthogonal to each other Shah et al. (2023). Consequently, for training the merged LoRA adaptor, the cosine similarity between merge vectors $m_o$ and $m_v$ is minimised as in **??**, making the columns of the weights of the view and object adaptors orthogonal to each other by disentangling them.

### A.3 EXTENDED DETAILS ON LOSS FUNCTIONS FOR LOW RANK MERGING

For low rank merging of the view and object LoRA, we focus on two aspects. First, we want to minimize the concept leak between the view and object LoRA which is defined by the cosine similarity between the columns of view and object LoRA. Second, we want to preserve the ability of the merged LoRA for independent generation of reference view concept and novel object concept by minimizing a L2 (mean squared loss) loss between view and object generated by the merged LoRA and the original view and object LoRA respectively as seen in Figure 2 in the main paper. In order to avoid superposition between two concepts, the cosine similarity is minimized between learnable merge vectors $m_v$ and $m_o$ for each layer. Let $L_{vo}$, $L_v$ and $L_o$ be the merged, view and object LoRAs respectively. Given these aspects, the loss function is defined as shown in Equation 4.

$$
\begin{aligned}
Loss_{vo} = {} & ||(D_\theta \oplus L_{vo})(\Phi_v, t_v) - (D_\theta \oplus L_v)(\Phi_v, t_v)||_2 \\
& + ||(D_\theta \oplus L_{vo})(\Phi_o, t_o) - (D_\theta \oplus L_o)(\Phi_o, t_o)||_2 \\
& + \lambda \sum_i |m_v^{(i)} \cdot m_o^{(i)}|
\end{aligned}
\tag{4}
$$

In the equation, $t_v$ and $t_o$ refer to the respective view text prompt and object text prompt. The update weight matrix for $L_{vo}$ is calculated as in Equation 3. $\lambda$ is a suitable multiplier for cosine-similarity loss term. The weights of the base stable diffusion model, $D_\theta$, and the individual LoRAs are kept frozen, so only the merge vectors are updated.

### A.4 MORE RESULTS ON DTU MVS AND DREAMBOOTH DATASET AND QUALITATIVE COMPARISON WITH BASELINES.

In this section we present additional qualitative results to demonstrate our method's transfer performance on DTU MVS and the DreamBooth Dataset. The results include view transfer from several different reference views to different objects of DTU and DreamBooth dataset unseen by our model. Refer to Figure 12 and Figure 13 respectively. Figure 14 shows qualitative comparisons with the current state-of-the-art NVS methods. We observe that AnyView generates images, visually faithful (column 1) to the original object samples. Furthermore, our method performs better in generating complex images in accurate view points (column 3,4) whereas ViewNeti Burgess et al. (2023) fails to preserve the structural integrity of the objects and Nerdi Deng et al. (2023) produces blurred or incomplete artificats. In addition to the DTU results, we also provide AnyView's performance on unique, in-the-wild images that the diffusion model likely does not have a strong prior for in Figure 15. We use the examples of personal figurines, specific statues and toys. AnyView, performs consistently for highly specific objects as well.

### A.5 IMPORTANCE OF VISUAL CUES FROM BACKGROUND

Considering that we are not provided with any 3D knowledge of the camera/object pose in our training, we defined the camera view as a visual concept is learnt from the cues in the background of the object. In this section we provide ablations in support this. We run the experiments in three settings as shown in Figure 16. First (**A**), we remove the background information from both the view and object training samples. In the second setting (**B**), we remove the background of the image sample for view training only, and in the third setting (**C**), the background is removed from the object samples, keeping only the background of the view samples. We observe that the absence of backgrounds during view and object training affects the view transfer to the object while merging. As there is no visual cue at all, it's difficult for the view model to learn the high level concept. Even after providing the background information to object training, the merge fails to capture a reliable view. This is a natural extension to the fact that view is heavily learnt from the view LoRA, which was not provided any spatial information from the background. In (**C**) however, when we provide this spatial information to the view LoRA, the merged results improve trying to replicate the original view. Interestingly, we further observe that having visual cues both in the view and object training stages captures a reliable view as well as a better structure of the novel object.

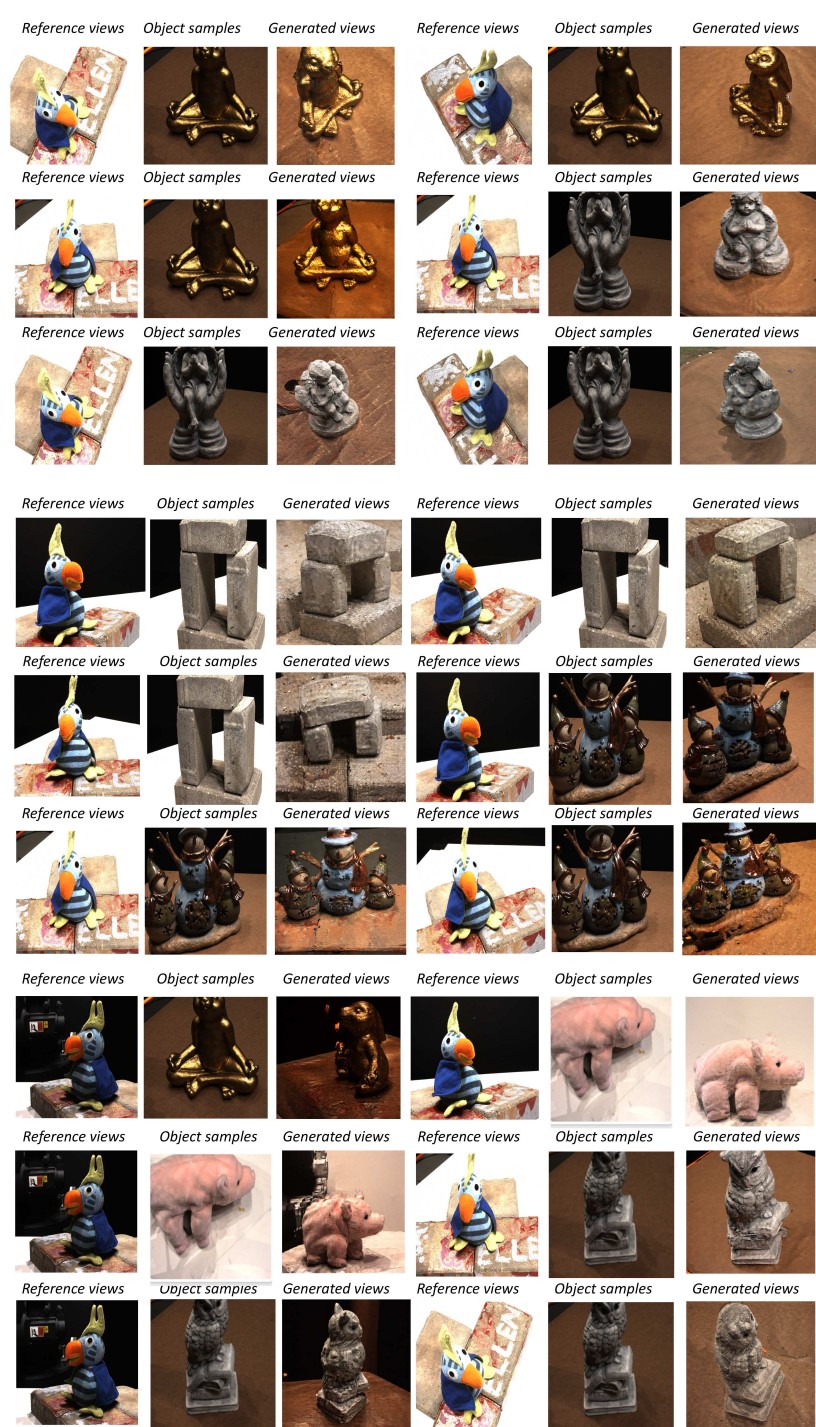

Figure 12: **View Transfer on DTU MVS Dataset.** We provide the reference view used for training the view LoRA and one sample image used in training the object LoRA. For each of this pair we provide the corresponding view generated by AnyView.

The importance of having background for view training is further highlighted in Figure 17. In this case, we try to transfer the top view of a chair to a dog, but we similarly remove the backgrounds from the view and object samples. We can see that the generations are heavily biased towards constructing the front view of the dog instead of the top view. While the view LoRA tries to learn

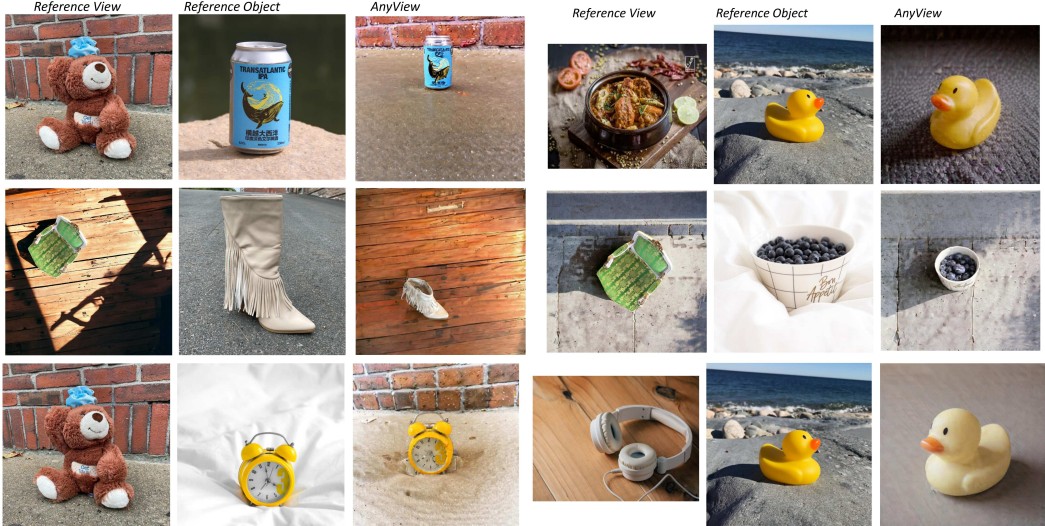

Figure 13: **View Transfer on DreamBooth Dataset.** We provide the reference view used for training the view LoRA and one sample image used in training the object LoRA. For each of this pair we provide the corresponding view generated by AnyView.

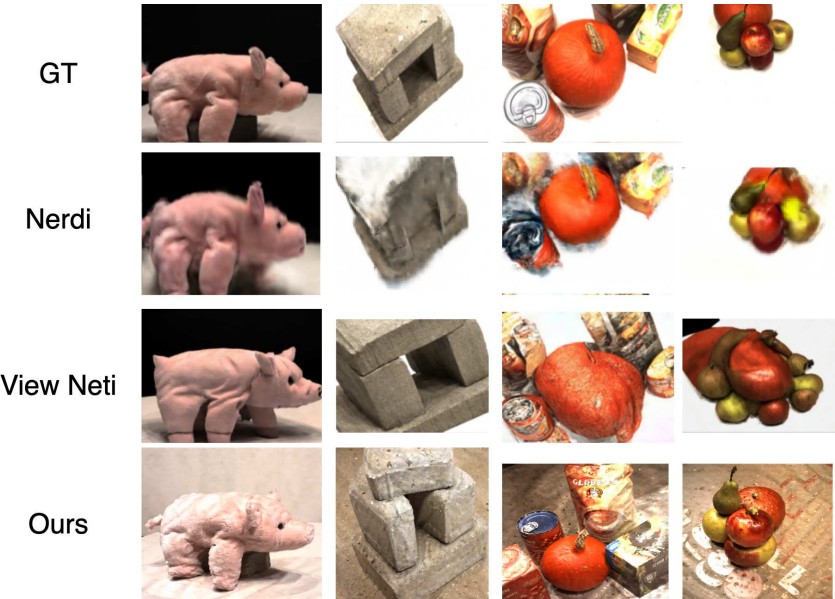

Figure 14: **Qualitative comparison of novel view synthesis on DTU.** Notably, as mentioned in the main text, ViewNeti has an competitive advantage over AnyView as it underwent prior training on the *full* training set of DTU MVS dataset. Similarly, NeRDi uses depth maps to regularize the 3D geometry, which were not available to AnyView. In spite of this, it can be observed that AnyView captures comparable, if not better, semantics, accurate to the original view points.

the concept (owing to the knowledge space of stable diffusion, which knows what a chair from top view looks like), we can see that only the last two samples in the bottom row of Figure 17 somewhat captures the view concept. However, with the presence of background cues in the reference view image, the transfer of the view concept is much more accurate and consistent as can be seen in Figure 4, Section 4.1, **Comparison on Datasets** of the main paper. Although AnyView struggles in absence of background cues or abstract background where it is very difficult to understand the

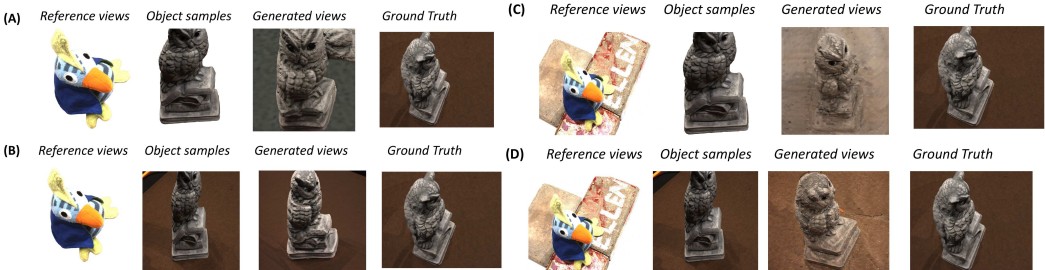

Figure 15: **View Transfer on Unique Objects.** We provide the reference view used for training the view LoRA and one sample image used in training the object LoRA. For each of this pair we provide the corresponding view generated by AnyView.

orientation in world space, it is quite adaptable to a wide range of backgrounds if there are sufficient details like object shadows, texture etc in the background. This is clearly observed in Figure 18.

Figure 16: **Effect of background in view transfer.** **(A)**: We remove the backgrounds from both the view and object samples. In **(D)**, we show the corresponding results obtained on having backgrounds in both the view and object training samples. The generated view in **(A)** is inconsistent in camera elevation to that of the reference view image, whereas the generated view in **(D)** (in presence of backgrounds) is closer to ground truth. **(B)**: We remove the background from the view training sample. The generated view captures the view better than in **(A)**. **(C)**: With only background in the view sample, the generated view captures the reference view well attributing to the fact that view concept is learnt from visual cues in background given in the reference view sample.

## A.6   ABOUT BACKGROUND IN MERGED RESULTS

The merged results inherit backgrounds from the view and/or object samples provided during training. However, we observe that it is not directly clear how to predict the amount of influence each has on the background of the final generated image. We attribute this to the training scheme for merging the view and object LoRAs. The merge vectors $m_v$ and $m_o$ are learnable parameters and hence the LoRA layers are being weighted in an adaptive manner. The $m_v$ and $m_o$ are learnt in a way such that while it reaches a optimal point which captures the view concept as well as maintains high fidelity to the novel object, the process also finds a compromise between the backgrounds. Furthermore, we apply no constraints on the background other than the fact that they share visual consistency to the view concept which is already taken care of by the in-painting abilities of stable diffusion Burgess et al. (2023). We conduct merge experiments in two different settings to observe the trend in $m_v$ and $m_o$. In setting **(A)**, we maintain similar backgrounds for both view and object samples and in **(B)**, the reference view background and object background are kept distinctly different as shown in Figure 19. We observe that the weighting vectors $m_o$ and $m_v$ are similar in **A** whereas in **B**, $m_v$ is more dominant than $m_o$. As a result, the view background is inherited in the merged samples as shown in Figure 19.

*No view background No object background*

*Reference view*       *Generated views*

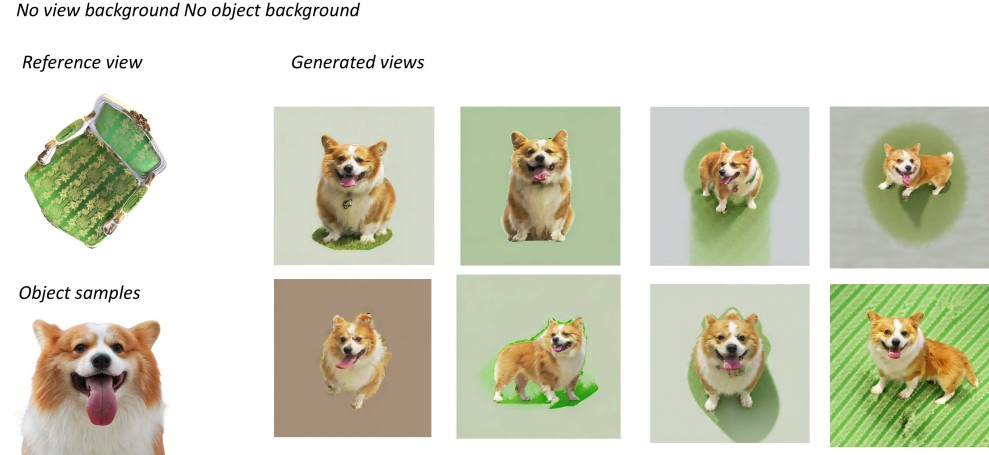

*Object samples*

Figure 17: **Generating top views of a novel object.** The views are generated under the condition that the object and view background are not given during respective trainings. As observed, the generated views are heavily biased towards generating a forward facing dog. Given the prior knowledge that the underlying diffusion model (SDXL) most likely has seen the top view of a chair, the last two images in the second row were able to orient itself to the concept but the transfer is clearly not reliable or consistent with the reference view (3/10 generations as opposed to 9/10 generations in presence of backgrounds).

*Reference View* ⟶

*Reference Object*

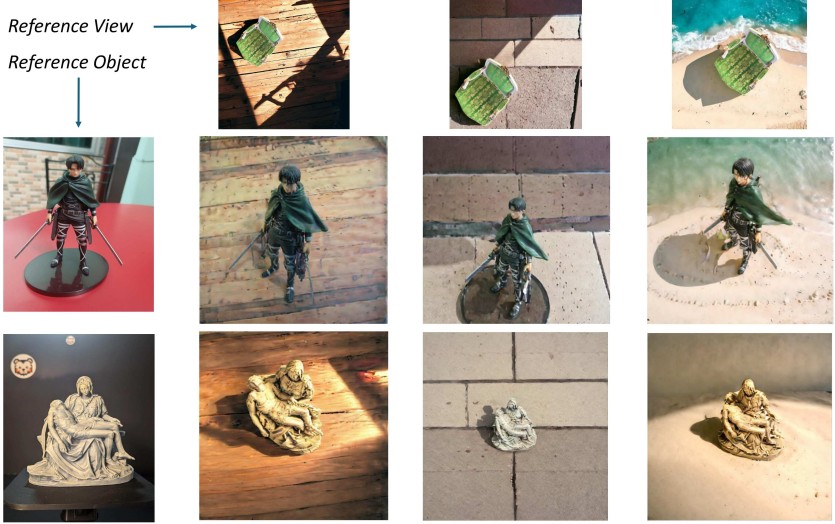

Figure 18: **Adaptability towards different background in reference views.** Three different background settings for the similar reference view is considered; a wooden floor, brick floor and beach. AnyView captures the reference view well in all three background settings.

## A.7 HANDLING OCCLUSIONS AND COMPLEXITY IN THE OBJECT IMAGES

Referring to Figure 20, in Row (I), no object sample shows the back of the dog, which is occluded in the front shots. The generated images reasonably hallucinate the back of the dog. Although these hallucinations may not precisely reflect the occluded region's exact visual composition, they

**(A)**
Reference views  Object samples  Generated views

**(B)**
Reference views  Object samples  Generated views

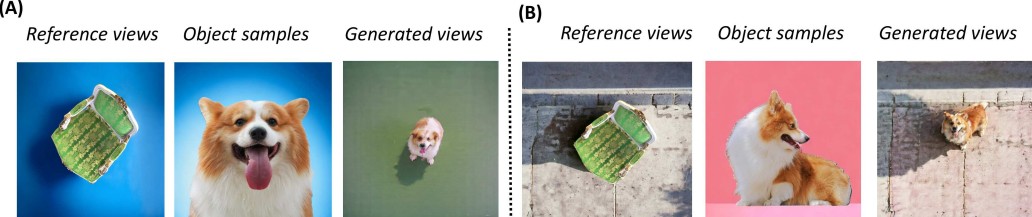

Figure 19: **The inherited backgrounds in views generated from the merged LoRA. A**: Similar backgrounds are maintained between the view and object samples. The generated views inherits a greenish-blue background, which is not the same as the input backgrounds. However, it is attributed to the optimal convergence of $m_v$ and $m_o$ learnable weighting vectors. **B**: Different backgrounds are provided to the view and object samples. The generated view inherits the background of the view. During training, we observe that the $m_v$ is optimised such that the view LoRA is weighted more than the object LoRA. Although its difficult to formalize a trend, the background bias is clearly dependent on the learnt $m_v$ and $m_o$. We expect to address this in future work so as to generate more predictable backgrounds.

are consistent with the visible object features. Similarly, in Row (II), none of the object samples fully reveal the statue's face. Yet, in subpart (a), the model provides a reasonable estimate of the statue's facial features, and in subpart (b), top of the head. Additionally, for the scenario where the

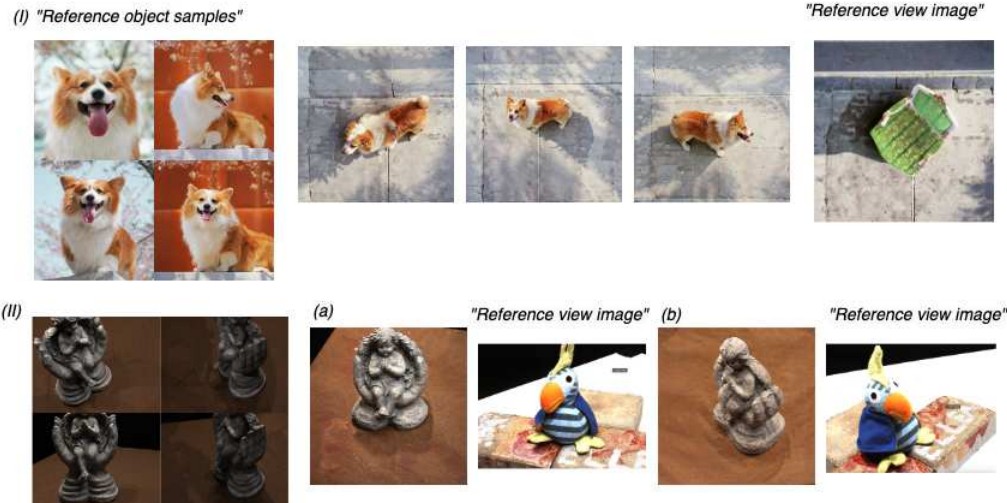

Figure 20: **AnyView hallucinating occluded regions in the object image samples.** In **Row A** we see that the back of the dog is occluded. Our method reasonably estimates the back of the dig in the views reconstructed that is consistent to the visual features of the dog. Similarly, in **Row B** FSViewFsuion provides an estimate of the occluded regions in the statue's face.

object to which the view is transferred is structurally more complex than the source object, we have already demonstrated via the view transfer from a simple chair to a complex TV show character in Section 4.2, Fig 9 in the main paper. However, when reconstructing multiple objects in various views, as shown in Figure 21, we find it notably more challenging than single object settings. For instance, in Row (A), Column 3, an additional bean appears, while in Row (C), Column 2, an apple is missing. This issue stems from the nature of DreamBooth finetuning, where a single unique token may represent multiple objects generically (e.g., using "groceries" for Row (A)). As a result, there could be additional or missing artifacts in the images. We also provide results on some controlled data samples as shown in Figure 22. In most of the samples, it is observed that the reference view transferred faithfully. However, the positional relation between the multi-objects may change given

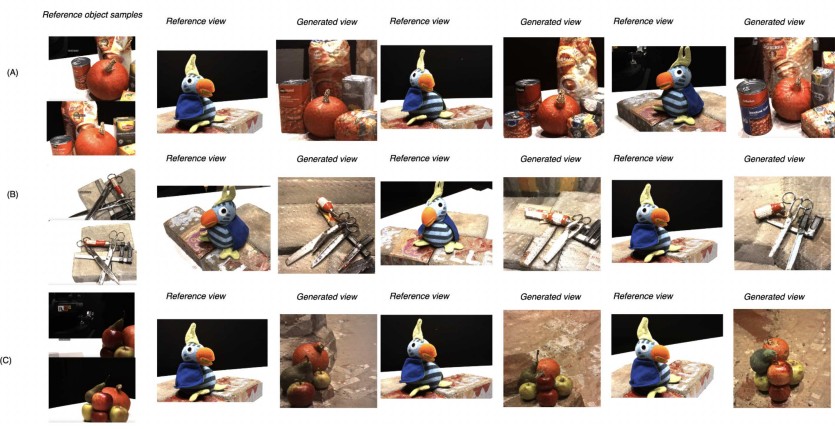

Figure 21: **Generation of multi-object images from various view points.** Although generating views of multi-object images are complex, our method still produces reliable object semantic in different view points as seen in coloumn 2,3 and 4.

the limitation with DreamBooth. Despite these challenges, AnyView still reliably generates images from various viewpoints.

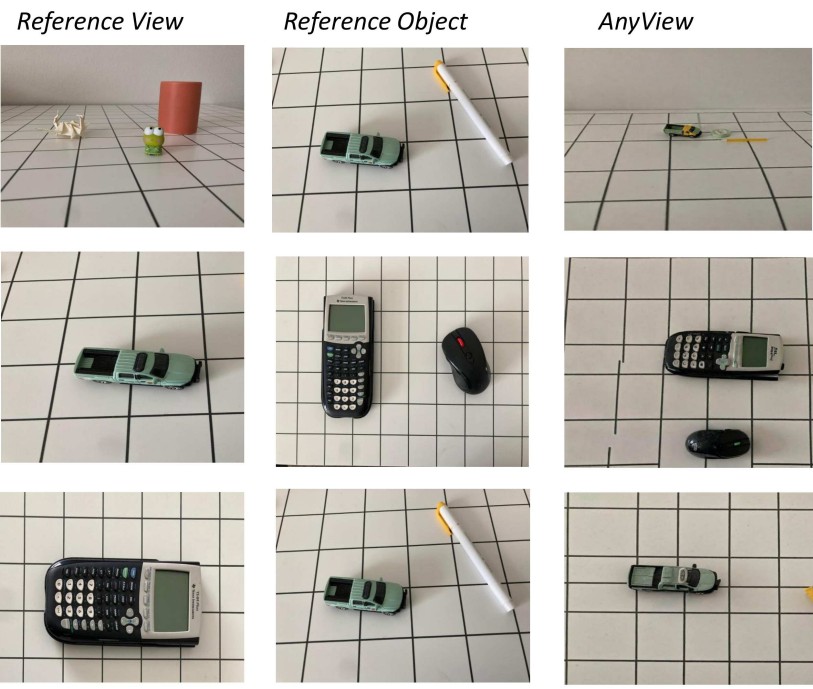

Figure 22: **Generation of multi-object images from various view points with controlled data samples.** Here we prepare some in the wild samples of multi-object images. **Row-1**: an additional duct-tape is present in the generated view. **Row-2**: The generated view captures the calculator and mouse relatively well **Row-3**: Both the car and the pen are captured in the reference view, however only part of the pen is visible.

