# OpenReview forum: "AnyView: Few Shot Personalized View Transfer"
_ICLR.cc/2025/Conference — Submitted to ICLR 2025_

### Official Review · Reviewer_xQX6 · 2024-10-23

**Soundness:** 2
**Presentation:** 2
**Contribution:** 2
**Rating:** 5
**Confidence:** 4

**Summary:**

The paper is about customized image synthesis of an object with a specific view point. They leverage the DreamBooth customization approach in order to learn object and view features. The model requires few-shot examples for custom object and view synthesis.

**Strengths:**

Based on the claims of the paper
1. The method does not require any prior 3D data, to synthesize a custom object with a specific view.
2. It is a few-shot approach, requiring 3-4 examples for object and only 1 example for view to synthesize the image.
3. They showcase the background features importance for view synthesis.

**Weaknesses:**

1. Weak scientific novelty. They just use the DreamBooth to learn 2 different concepts leveraging the LoRA method for it.

2. Limited quantitative and qualitative results. The authors provide results using only 1 dataset (DTU MVS), which is not enough to prove the claims. Also, they can be cherry picked, especially, when the authors manually estimate the transformation of competitor model (Zero123) for the comparison.

3. Not convinced with the claim of disentangling the view synthesis from the concept. Not enough evidence are shown in the paper. The model is able to learn concepts (whether they are style, object, or scenes). There is a more chance that it attempts to learn the scene, not the view. That is why in almost all results, any image synthesized from the top camera view, object is faced to the camera, while, the reference object does not have that position. Also, the background is always changed in synthesized images (probably because of LoRA), which works against the claim of view disentanglment.

4. Poor qualitative results. The synthesized images are not appealing qualitatively (Fig 5, Fig 12). The customized objects have poor fidelity.

5. Lack of details how the training is done, how the quantitative metrics are computed.

**Questions:**

I am willing to reconsider my review if authors address the weaknesses mentioned above and convince me otherwise.

---

### Official Review · Reviewer_HaBK · 2024-11-03

**Soundness:** 3
**Presentation:** 3
**Contribution:** 3
**Rating:** 6
**Confidence:** 3

**Summary:**

The paper introduces a method for generating personalized view transfers in image synthesis using diffusion models through the DreamBooth setup combined with low-rank adaptation (LoRA). More specifically, it proposes AnyView, a system that disentangles "view" as an independent high-level concept and leverage the personalization/transfer-learning algorithm of Dreambooth to extract this concept. Experiments on DTU and DreamBooth dataset validate its performance.

**Strengths:**

1. The paper brings forward a novel approach by defining the concept of "view" in image synthesis. By the Dreambooth personalization algorithm, the method requires minimal 2D data and no 3D information to extract such concept.
2. The experimental setup is robust, with evaluations on datasets such as DTU and comparisons some benchmarks, demonstrating AnyView's performance.
3. The paper provides a comprehensive analysis to give insight on how the setup like LoRA, background, complex object would affect the approach.

**Weaknesses:**

1. Evaluation Metrics. My strongest concern is the use of evaluation metrics to assess the method's performance. While traditional metrics like SSIM and LPIPS are presented, they may not entirely capture the visual quality of the view-transfered novel images. Incorporating perceptual metrics or human study evaluations is critical for this few-shot synthesis technique, which can assess the fine details of the generated images.
2. Dependence on Background Cues. While the model effectively generates personalized views, it relies significantly on spatial cues from backgrounds, as discussed in Sec. A.5. This dependence may limit its adaptability to varying or abstract backgrounds.
3. Complex Multi-Object Scenes. While the paper acknowledges that the method faces challenge in maintaining view consistency and object separation for the multi-object scenes, more experimentation on such cases could further validate the method's efficacy and potential modifications to address this complexity.

**Questions:**

Please refer to the weakness section.

---

> ### Comment · Reviewer_HaBK · 2024-11-29
>
> Thanks the reviewer for addressing the issues, I'll remain my rating.

---

### Official Review · Reviewer_TD66 · 2024-11-04

**Soundness:** 3
**Presentation:** 3
**Contribution:** 3
**Rating:** 6
**Confidence:** 4

**Summary:**

The paper presents a novel approach for enhancing personalization in visual tasks through a method that integrates personalization and Low-Rank Adaptation (LoRA) fine-tuning for view transfer. The authors address the "forgetting" problem prevalent in personalization models when learning multiple IDs simultaneously, providing a clear motivation for their proposed architecture. Through thorough experimentation, the method demonstrates competitive performance results against various baselines, including methods that have been trained on more data. The findings are substantiated across multiple evaluation metrics.

**Strengths:**

- The paper is clearly articulated, with a well-defined methodology supported by diagrams, effectively motivating the proposed architecture and addressing the "forgetting" problem in personalization models.
- The method demonstrates notable performance improvements over comparable baselines, achieving competitive results against specialized novel view synthesis (NVS) methods despite relying on a simpler approach with fewer input images. The authors provide comprehensive results across various metrics, including SSIM, PSNR, and LPIPS, highlighting the effectiveness of their approach in diverse scenarios.
- The authors provide a thorough analysis of design choices, including the impact of background in training data and object complexity on the method’s performance.
- The originality of the approach is noteworthy, particularly in its application of personalization and LoRA fine-tuning for view transfer tasks.

**Weaknesses:**

- **Evaluation Metrics:** The evaluation method used, particularly regarding masking, are unclear and warrant further explanation.
- **Qualitative Results:** Some views selected for qualitative results appear standard, especially in Figure 5 with Zero-1-to-2 comparison visualizations. Additionally, there is a lack of qualitative results for other baselines- although some are presented in Figure 13 in the appendix, they are not super convincing as Anyview results seem to lack fine-grained viewpoint consistency in view synthesis, while also lacking some fine-grained detailed in complex tasks.
- **Background Sensitivity:** The performance of the method is inconsistent, particularly sensitive to background context. Structured backgrounds (e.g., forests, tables) yield more reliable results than uniform backgrounds (e.g., grass). This sensitivity suggests the model may learn unintended spatial cues, potentially hindering its generalizability across diverse environments and highlighting a brittle understanding of 3D relationships.
- **Evaluation Gaps:** The paper lacks an analysis of failure cases and their characteristics, which is critical for understanding the limitations of the approach.
- **Technical Limitations:** There is no clear strategy for managing multiple objects or complex scenes within the proposed framework.

**Questions:**

1. What is the precise definition of "masking" in the context of your evaluation methodology?
2. How is the view generated from a single image, considering that views are inherently relative? Is it relying on the semantic prior of "canonical pose" of diffusion models? Would incorporating multiple images of the same view but with different objects improve performance? Additionally, how does the method address viewpoint ambiguity?
3. What does Figure 7 represent? Are all the images inference attempts to generate an ideal view? The inconsistencies in these images raise questions regarding their interpretation.
4. What specific failure cases or limitations can you identify in your approach?
5. Is it feasible to extend this method to effectively handle multi-object scenes, and if so, how would that be accomplished?

---

### Official Review · Reviewer_Wsxd · 2024-11-04

**Soundness:** 2
**Presentation:** 2
**Contribution:** 2
**Rating:** 5
**Confidence:** 5

**Summary:**

The paper introduces AnyView, a method for transferring specific viewpoints to novel objects using few-shot learning with diffusion models. Building upon DreamBooth, the authors demonstrate that a pretrained stable diffusion model can learn the high-level concept of a view from a single image without relying on explicit 3D priors. They use Low-Rank Adaptation (LoRA) to separately learn the view and object concepts and then merge them to generate images of the novel object from the desired viewpoint. Experiments on the DTU dataset and natural images show that AnyView can efficiently generate reliable view samples, outperforming several methods in certain metrics.

**Strengths:**

- This work studies a new task, learning a view from a single image.
- The authors provide evidence that diffusion models can learn and transfer high-level concepts like views, which could have broader implications for generative modeling.

**Weaknesses:**

- The method may struggle with complex scenes involving multiple objects or significant occlusions, as noted in the limitations.
- The results shown in experiments are all simple or celebrities, which SDXL model is highly possible to have priors. Using some unique objects will make the method more convincing.
- The reference images of rare object shown in appendix already have diverse views, some are even very similar to the view to learn. This undermines the effectiveness of the method.
- The method is simple combination of DreamBooth and ZipLoRA, which has limited innovation.
- The figures in the paper are very low-resolution and unlcear, making it difficult to read.

**Questions:**

See weaknesses above.

---

> ### Comment · Reviewer_Wsxd · 2024-11-27
> **Response to Authors**
>
> Sorry, to make it clear, I mean the rows in your new Fig12 part 2. The pig toy images seem to be obvious failure for me.
> Besides, if authors would like to address the effectiveness of this method, it's lack of quantitative metrics to evaluate. I think it's better to define the orientation or coordinates of both the view and the object to give some quantitative measurement.

---

### Meta-Review · Area_Chair_gqbi · 2024-12-10

**Metareview:**

The paper receives mixed scores from the reviewers. While reviewers appreciate the interesting task and comprehensive analysis, they also find the method with incremental novelty, the evidence poor, and the dataset limited, which are not fully addressed in the rebuttal. The authors are encouraged to address these comments in the revised version.

**Additional Comments On Reviewer Discussion:**

During discussion, reviewers all mentioned that the rebuttal is not satisfactory and they are not fully convinced. More solid experiments are needed to justify the proposed method. Reviewer TD66 also mentioned that controllable background generation is a problem.

---

### Decision · Program_Chairs · 2025-01-22

Reject